# A community pharmacist-led smoking cessation intervention using a smartphone app (PharmQuit): A randomized controlled trial

Narong Asayut[1☯], Phayom Sookaneknun Olson[2☯]*, Juntip Kanjanasilp[3☯], Preut Thanarat[4‡], Bhattaraporn Senkraigul[4‡], Chuthathip Sittisarn[2☯], Suratsawatee Suksawat[2☯]

1 Doctor of Philosophy in Pharmacy Program, Faculty of Pharmacy, Mahasarakham University, Khamriang Sub-District, Kantarawichai District, Maha Sarakham Province, Thailand, 2 The International Primary Care Practice Research Unit, Faculty of Pharmacy, Mahasarakham University, Khamrieng Sub-District, Kantarawichai District, Maha Sarakham Province, Thailand, 3 The Clinical Pharmacy Research Unit, Faculty of Pharmacy, Mahasarakham University, Khamrieng Sub-District, Kantarawichai District, Maha Sarakham Province, Thailand, 4 Faculty of Informatics, Mahasarakham University, Khamrieng Sub-District, Kantarawichai District, Maha Sarakham Province, Thailand

☯ These authors contributed equally to this work.
‡ PT and BS also contributed equally to this work.
* phayom.s@msu.ac.th

**Data Availability Statement:** All relevant data are within the manuscript and its Supporting Information files.

## Abstract

WHO supports the harnessing of mobile technologies to improve access to smoking cessation services. As such, this study evaluated the effectiveness of smoking cessation services provided by community pharmacists using PharmQuit compared with standard care. The study was a prospective, multicenter, randomized controlled trial that included 156 participants who were 18 years or older and smoked at least one cigarette daily for a month, were ready to quit, willing to participate, and had a smartphone. The study was performed at seven community pharmacies in three provinces in Thailand. Participants were allocated to the intervention (n = 78) and control groups (n = 78). Both groups received the usual smoking cessation services with pharmacotherapy and counseling from community pharmacists for 6 months. The intervention group received PharmQuit as an additional service. Both groups were scheduled for follow-up visits on days 7, 14, 30, 60, 120, and 180. The primary outcome was continuous abstinence rate on day 180. The secondary outcomes included 7-day point abstinence rate, number of cigarettes smoked per day, exhaled carbon monoxide levels, adherence rate to the program, and satisfaction with PharmQuit. An analysis using the intent-to-treat principle was performed. Smoking cessation rates and the number of cigarettes smoked per day were significantly higher during the follow-up visits in both groups (p < 0.05). However, there were no statistically significant differences between the two groups. The adherence rate to the smoking cessation program was higher in the intervention group than in the control group (74 days vs. 60 days, p > 0.05). The results showed the benefits of the contribution of community pharmacists. Although the inclusion of PharmQuit did not

**Funding:** We would also want to express our gratitude for funding supports from the Health Security Office region 7 Khon Kaen, the Smoke Free Pharmacy Network, Community Pharmacy Foundation, Thai Health Promotion Foundation, and the Faculty of Pharmacy, Mahasarakham University.

**Competing interests:** The authors have declared that no competing interests exist.

yield better results than pharmacists' counselling alone, it may help obtain better adherence to smoking cessation programs.

**Trial registration**: Thai Clinical Trials Registry: TCTR20200925004 on September 25, 2020 –retrospectively registered, http://www.clinicaltrials.in.th/index.php?tp=regtrials&menu=trialsearch&smenu=fulltext&task=search&task2=view1&id=6841.

## Introduction

Tobacco smoking is a major cause of premature death worldwide [1], leading to the death of up to half of its users. In 2017, smoking was the second leading risk (following hypertension) factor for premature death and disability globally, ranked by disability-adjusted life years (DALYs) [2]. The World Health Organization (WHO) has a global action plan to reduce the prevalence of tobacco use in persons aged 15 years and older by 30%, by the year 2025 [3]. In Thailand, the smoking prevalence among the general population in 2010 was 42%. With the combination of tobacco control policies and the rate of smoking cessation, the relative prevalence is predicted to be 34% in 2025, which is still higher than the WHO target [4]. Therefore, increased efforts to control tobacco use are essential for reducing the burden of non-communicable diseases in Thailand [4].

Counseling for smoking cessation is effective at helping smokers with quitting. Lancaster and Stead showed that different models of counseling provide benefits to participants [5]. Nicotine replacement therapy (NRT) increased the rate of quitting by 50–60% for people who attempted to quit, regardless of the setting [6]. A combination of pharmacotherapy and high-intensity behavioral treatment was found to be better than high-intensity behavioral treatment alone [7], and health professionals, such as pharmacists, are in a unique position to help smokers quit. Several systematic reviews have shown that pharmacist-led interventions result in better abstinence rates in smokers [8–11] and may also be cost-effective [11, 12]. However, pharmacy counseling programs still have a high dropout rate [13].

Although various mobile apps are available to help smokers quit, studies have shown that only two out of 50 apps were accompanied with scientific and professional support [14], and most apps were not customized to the users' needs [15]. An objective of this study was to help smokers adhere to a smoking cessation program, in which pharmacists assist and provide information and support for smokers. Developed for iOS and Android phones, PharmQuit aims to help "ready-to-quit" smokers with the assistance of community pharmacists. PharmQuit was developed using the five-user experience framework from the perspectives of smokers and pharmacists and was designed to deliver easy access to users. PharmQuit sends encouraging messages to the users' phone every day to remind them to keep a daily record of smoking and maintain abstinence. It also provides information on cravings and adverse drug reactions from medications. Users can see clinical data screened by their pharmacists in PharmQuit, and they are also able to send messages directly to their pharmacists through the Line@ option. In addition, they can see how well they are doing based on the avatar. Participants can also share their status with others in the PharmQuit community.

Current evidence confirms the benefits of mobile phone-based smoking cessation interventions on long-term outcomes [16]. Although most app studies were designed for self-use [17, 18], one study featuring collaboration between smokers and physicians through an app and web-based system yielded positive results. However, only a few studies on smoking cessation apps have been conducted in community pharmacies and those were evaluated for short-term

(8 weeks) outcomes [19, 20]. The aim of this study was to evaluate the effectiveness of the pharmacist-led smoking cessation app, PharmQuit, compared to a control group without the app. The objectives included evaluating the primary outcome of the continuous abstinence rate (CAR) at 6 months and the secondary outcomes of 7-day point abstinence rate (PAR), number of cigarettes smoked per day, exhaled carbon monoxide level at every visit, adherence to the smoking cessation program, and satisfaction with the app at 6 months.

## Materials and methods

### Design

This was a multi-center study using an open-label randomized trial with a control group in seven community pharmacies in three provinces in Thailand. Stratified random sampling was used for both the control and intervention groups and was based on age, sex, and nicotine dependence. The study adhered to the CONSORT guidelines [21] and includes a completed CONSORT checklist in S1 File. Cash compensation of $1.50 US dollars per visit was given to each smoker and $16.68 US dollars per smoker was given to each community pharmacist.

**Ethical approval and consent to participate.** The study protocol (S2 File), consent forms, and tools (S3 File) were approved by Mahasarakham University (ID: 033/2559). The certificate of approval is in the S1 File. The participants provided written informed consent to participate in the study. This trial was registered retrospectively in the Thai Clinical Trials Registry (TCTR20200925004) on September 25, 2020. The researchers were unaware of the requirement to register the clinical trial prior to the enrolment of participants, but amendments were made to our institutional training materials to reduce the likelihood of this error occurring in the future.

**Participants.** The recruitment was through invitations by pharmacy students, community pharmacists, health care providers, and health care volunteers. The recruitment period was July 30, 2017, to August 28, 2018. The study was conducted in January 2019. Eligible participants were smokers who were (1) aged 18 years or older; (2) smoking at least one cigarette per day for a month or more; (3) ready to quit smoking or in the preparation stage; (4) willing to participate in the study; (5) able to complete self-recording; and (6) had a smartphone. The exclusion criteria were women who were pregnant or breastfeeding, individuals with cardiovascular disease, and individuals currently enrolled in another smoking cessation program. The sample size was estimated based on two independent proportions of equal samples [22]. The quit rate after 6 months of treatment (13.8%) and the control (1.3%) groups were based on a 2010 study by Burford et al. [23]. The minimum sample size was 69 people per group (with $\alpha = 0.05$, two-tailed) with 80% power to reject the null hypothesis in the quit rate at the 6-month follow-up. A dropout rate of 15% [24] was estimated; thus, the sample size was increased to approximately 80 smokers per group.

**Randomization.** Stratified random allocation was used to achieve equal assignment to the two groups. Stratification was performed according to three factors: sex, nicotine dependence level as determined by the Fagerström test (FTND) score [25], and age. A computer-generated random sequence determined by a chance process and could not be predicted was used to assign participants to the intervention and control groups (1:1). A printed randomized table was delivered to each pharmacy by a researcher (S2 File). Pharmacists and participants were not blinded. The pharmacists enrolled the participants, then, using a randomized table provided by the researcher, allocated participants to either the intervention group or the control group. The main researcher (Asayut N) assessed the allocation procedure of each pharmacy by checking the registered date and entry sequence in the web system compared with the randomized table from each pharmacy.

**Intervention group.** Smokers assigned to the intervention group met one-on-one with a community pharmacist at the respective pharmacy. The duration of the first visit was approximately 30 min. The pharmacists asked if the participants were willing to quit smoking, if they agreed to participate in the study, and to complete a consent form. The pharmacists reassured the participants that choosing to quit was the best decision and emphasized the benefits of quitting. The pharmacists interviewed the participants for general information, intention to quit, struggles in quitting, history of attempting to quit, smoking habits, and nicotine dependence as determined by the Fagerström test. Blood pressure, weight, and exhaled carbon monoxide (CO) determined by a smokerlyzer, were measured by pharmacists. Following the treatment plans, the pharmacist checked the randomized table to see if the participant was in the intervention group and registered their name to the web system (http://www.smokefreerx. com/) to get a username for the participant login. PharmQuit was then introduced and registered on the participant's mobile phone. At the end of the service, the pharmacist scheduled the next visit.

Nicotine gum, nortriptyline, sodium nitrate 0.5% mouth wash, *Vernonia cinerea* lozenges, and herbal medicine were provided as options for participants who had FTND scores of at least 4, were smoking at least 10 cigarettes per day, or had a history of failure to quit smoking. Contraindications were checked prior to dispensing. The pharmacists counseled the participants on the following: drug name, dose, regimen, administration, duration of therapy, adverse effects, and the disposal of nicotine gum. Pharmacists dispensed medications following the smoking cessation practice guidelines of Thailand [26].

Follow-up visits were scheduled for days 7, 14, 30, 60, 120, and 180. If the participants did not adhere to the schedule, follow-up was conducted by telephone, Line app, or Facebook messenger. Each follow-up visit took approximately 10–20 minutes. The pharmacists assessed smoking status, adherence to medication, adverse drug reactions, PharmQuit use, and the participants' overall status, to evaluate obstacles and provide encouragement. The pharmacists encouraged the participants to continue in the cessation program and did not blame them if progress had not been made. Blood pressure, weight, and exhaled CO were also recorded during follow-up sessions. At the end of the appointment, the pharmacists refilled the medications and scheduled the next visit.

**Control group.** Smokers assigned to the control group met one-on-one with a community pharmacist at the community pharmacy. The participants received the usual pharmacy services on smoking cessation. The procedure for the control group was as previously described for the intervention group; however, these participants did not have access to PharmQuit.

**Outcomes.** The pharmacists assessed outcomes at every visit. The primary outcome was the proportion of participants who remained abstinent continuously for 6 months (CAR). Secondary outcomes were the proportion of patients who remained abstinent for at least seven days before each visit (7-day PAR), the proportion who exhaled less than 7 ppm of CO, the proportion who could adhere to the follow-up schedule, the proportion of adverse events during the study, the number of cigarettes smoked per day, and the average score of satisfaction with PharmQuit, which was evaluated only in the intervention group on day 180.

The satisfaction questionnaire for PharmQuit was specifically developed for this study using a 5-Likert scale. Each of the five items was rated as 1 (very unsatisfied), 2 (unsatisfied), 3 (neutral), 4 (satisfied), and 5 (very satisfied). The Cronbach's alpha was 0.923. The user experience theory [27] was used as a framework for developing the questionnaire. The validity was verified by three experts in questionnaire development and research. There were 21 questions in five dimensions: objectives in quitting smoking (five questions with 25 scores), scope of application (four questions with 20 scores), format and interaction (four questions with 20

scores), design (four questions with 20 scores), and appearance (four questions with 20 scores). Online and paper-based questionnaires were administered to the participants in the intervention group at the follow-up visit on day 180.

**Data analysis.** Demographic data were compared between the intervention and control groups. Descriptive statistics are presented for the baseline as shown in Table 1. Categorical variables were compared using the chi-squared test, and continuous data were compared using the independent t-test for normally distributed data and the Mann-Whitney U test for data that with non-normal distribution. Intention-to-treat was applied in all analyses using the advantage of all available data points. Primary and secondary outcome variables were checked for distribution, outliers, and missing patterns. If participants missed in between visits, the information from the previous visit was assumed; for example, a participant came on day 14 with smoking then on day 30 with no smoking, day 60 would be recorded as no smoking. The missing data in the ratio scale were imputed by multivariate imputation by chained equations using R program version 4.1.1.

The primary outcome, CAR, was defined as participants who started abstinence from day 7 and remained abstinent at day 180. The 7-day point abstinence rate was defined as participants who could be abstinent for at least seven days before each visit. Differences in CAR and 7-day PAR between the groups were assessed using logistic regression. The odds ratios (ORs) and 95% confidence intervals (CIs) were calculated. The number of cigarettes smoked per day was

**Table 1. Baseline characteristics of participants in the intervention and control groups.**

| Characteristics | Intervention group No (%) (n = 78) | Control group No (%) (n = 78) | p-value |
|---|---|---|---|
| Gender: male | 71 (91.0) | 72 (92.3) | 0.772[b] |
| Age (years, mean ± SD) | 33.5±14.2 | 35.0±16.4 | 0.532[a] |
| Weight (Kg.) (mean±SD) | 69.5±14.8 | 70.2±15.0 | 0.771[a] |
| Blood pressure (mmHg) | | | |
| SBP (mean±SD) | 125.1±16.3 | 129.4±17.0 | 0.109[a] |
| DBP (mean±SD) | 77.2±10.3 | 78.9±11.2 | 0.391[a] |
| Having underlying disease | 24 (30.8) | 25 (32.1) | 0.908[b] |
| - Diabetes | 3 (12.5) | 8 (32.0) | |
| - Hypertension and ischemic heart disease | 4 (16.7) | 2 (8.0) | |
| - Asthma | 3 (12.5) | 5 (20.0) | |
| - Others (allergy, GI, pain, depression, dyspepsia, GERD) | 14 (58.3) | 10 (40.0) | |
| Length of time as a smoker (months) (mean±SD) | 183.8±165.8 | 191.9±177.5 | 0.770[a] |
| Number of cigarettes smoked per day (mean±SD) | 11.9±6.9 | 12.3±8.2 | 0.767[a] |
| FTND score (mean±SD) | 3.7±2.4 | 3.5±2.6 | 0.588[a] |
| - Score 7–10: high nicotine addiction 3 | 13 (16.7) | 11 (14.1) | 0.896[b] |
| - Score 4–6: moderate nicotine addiction 2 | 28 (35.9) | 28 (35.9) | |
| - Score < 4: low nicotine addiction 1 | 37 (47.4) | 39 (50.0) | |
| Using medications for cessation | 40 (51.3) | 44 (56.4) | 0.521[b] |
| - Nicotine gum (with lozenge) | 18 (45.0) | 14 (31.8) | |
| - Lozenge | 9 (22.5) | 14 (31.8) | |
| - Nortriptyline (with lozenge) | 6 (15.0) | 8(18.2) | |
| - Nicotine gum and nortriptyline (with lozenge) | 2 (5.0) | 5 (11.3) | |
| - others (mouth wash, lozenge with mouth) | 5 (12.5) | 3 (6.8) | |

[a] Independent t test

[b] Chi-square, FTND stands for Fagerström test nicotine dependence.

Lozenge means *Vernonia cinerea* lozenge, mouth wash means 0.5% sodium nitrite solution.

presented as means ± standard deviation (SD). Linear regression was used to compare the groups, and mixed models were used for comparisons within each group. Estimates of effect using ORs with a 95% CI and p-values, were analyzed using logistic regression. SPSS version 19 was used for the analysis. All tests were two-sided, and α was set at 5%. This study was open-label and the researcher (Olson PS) who performed the analysis was blinded to the data.

## Results

We randomized a total of 156 participants to the intervention group (n = 78) or the control group (n = 78). Completion on day 180 was 30.8% (24/78) in the intervention group and 23.1% (18/78) in the control group (Fig 1). Failure to follow-up was due to participants' unavailability, inaccessibility (after more than three attempts at contact in one week), and a loss of interest.

### Participant characteristics

In both groups, most of the participants were men. Participants in the control group were longer-term smokers than those in the intervention group. There was a lower proportion of high nicotine addiction in the control group than in the intervention group. Nevertheless, there were no significant differences in any of the baseline characteristics between the two groups (Table 1).

**Clinical outcomes.** The CAR at 180 days was similar in both groups, as shown in Table 2. At the end of the study, 25 participants stopped smoking in the intervention group; however, only 11.5% abstained for 180 days. It was discouraging to see one participant who had been abstinent from day 7 return to smoking on day 180. In the control group, there were 27 participants who stopped at the end of the study; however, only 12.8% abstained for 180 days. Most of the participants who returned to smoking could not stop smoking at the end of the study, with the exception of two participants in the control group who returned to smoking on day 30, and then abstained from day 60 to day 180, as shown in Table 3.

The 7 day-point abstinence rates were not different between the groups at each follow-up visit. A comparison within the groups revealed significantly higher abstinence rates from day 7 to day 180 (p < 0.05), as shown in Table 3. The number of cigarettes smoked per day was not significantly different between the groups. However, the number of cigarettes smoked per day on day 180 decreased significantly when compared with that on days 0, 7, 14, 30, 60, and 120 in the intervention group (p < 0.05) and days 0, 7, 14, and 30 in the control group (p < 0.05), as shown in Table 4. There was no significant difference in exhaled CO between both groups at any visit, but the results showed improvement in exhaled CO from day 0 to day 180 in each group (p < 0.05), as shown in Table 5. Of the 29 participants in the intervention group who accessed PharmQuit consistently, 12 (41.3%) were successful at quitting and 17 (58.6%) failed to quit smoking.

Medication for cessation was used in both groups. Eleven participants (27.5%) in the intervention group and 16 participants (36.4%) in the control group reported at least one adverse event. Most of the adverse events included dry mouth and throat (47.4% in the intervention group and 50% in the control group) as shown in Table 6. All adverse events were considered mild, and no referral to a hospital was reported during the study.

**Adherence to the smoking cessation program.** Adherence to the smoking cessation program in both the intervention and control groups was assessed. Table 7 shows that adherence was higher in the intervention group than in the control group for both the number of visits and days adhered to the cessation program; however, there was no significant difference between the two groups.

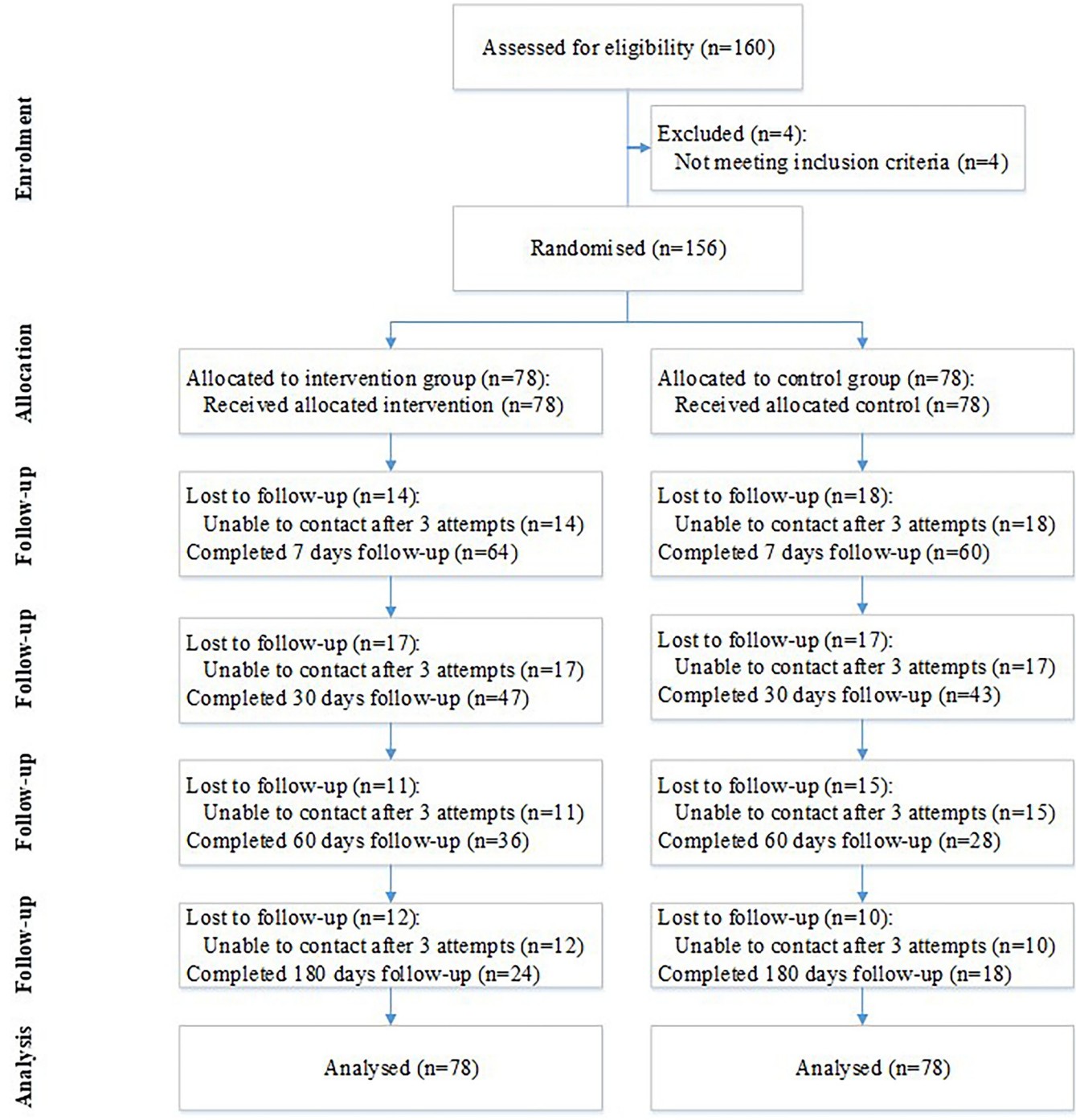

**Fig 1. Participant flowchart.**

Of the 78 participants in the intervention group, 37.2% used PharmQuit in month one. After 6 months, only 2.6% were using PharmQuit, as shown in Fig 2. The number of times PharmQuit was accessed was highest during the 1st month, which decreased over time. On day 180, 24 participants completed the 6-month follow-up. Fifty-four participants were unable to make contact. Thus, only 14 out of 78 participants (response rate: 17.9%) returned the

**Table 2. Continuous abstinence rate at 6 months between the intervention and control groups.**

| Outcomes at 6 months | Intervention group (n = 78) No (%) | Control group (n = 78) No (%) | OR | 95%CI | p-value[a] |
|---|---|---|---|---|---|
| 7 days abstinence rate | 25(32.1) | 27 (34.6) | 0.89 | 0.46–1.73 | 0.734 |
| 30 days abstinence rate | 24 (30.8) | 25 (32.1) | 0.94 | 0.48–1.85 | 0.863 |
| 60 days abstinence rate | 24 (30.8) | 25 (32.1) | 0.94 | 0.48–1.85 | 0.863 |
| 120 days abstinence rate | 21 (26.9) | 25 (32.1) | 0.78 | 0.39–1.56 | 0.483 |
| Continuous abstinence rate | 9 (11.5) | 10 (12.8) | 0.88 | 0.34–2.32 | 0.807 |

[a] Comparing between groups by logistic regression.

PharmQuit satisfaction questionnaire. They rated it highest for the three categories: font and size (median of 5, range 2), attractiveness and usability (median of 4.5, range 3), and the progress feature (median 4.5, range 2). Overall, satisfaction with PharmQuit was high for all questions. The convenience and ease of use of PharmQuit showed the lowest range of 1 as shown in Table 8.

## Discussion

Participants in both groups benefited significantly from the smoking cessation program provided by community pharmacists. Although there were no significant differences between the intervention and control groups, participants in both groups showed improvement in abstinence rate, number of cigarettes smoked per day, exhaled CO, and adherence to the cessation program. Adherence to the cessation program was not different between groups. However, the number of visits and days adhered to the program were higher in the intervention group than in the control group. This may be due to the motivation provided through PharmQuit, as the participants in the intervention group were highly satisfied with the app.

In this study, the CAR and 7-day PAR were lower in the intervention group than in the control group during most follow-up visits, but this difference was not statistically significant. Findings from other studies using different mobile apps are inconsistent. Other smoking cessation apps failed to highlight that each study uses a very different protocol, which may be the reason for the large discrepancies in outcomes. For example, in a study by Herbec et al. that included 300 community pharmacies in the UK, the results after 8 weeks showed a quit rate of

**Table 3. Comparisons of 7 day-point abstinence (PAR) rates between visits between the intervention and control groups on days 7, 14, 30, 60, 120, and 180.**

| Visit | Intervention group (n = 78) 7-day PAR No (%) | Status of quit attempts | p-value[a] | Control group (n = 78) 7-day PAR No (%) | Status of quit attempts | p-value[a] | OR | 95%CI | p-value[b] |
|---|---|---|---|---|---|---|---|---|---|
| Day 7 | 11 (14.1) | 11N | 0.001 | 12 (15.4) | 12N | <0.001 | 0.90 | 0.37–2.19 | 0.821 |
| Day 14 | 18 (23.1) | 7N | 0.065 | 21 (26.9) | 9N | 0.070 | 0.81 | 0.39–1.68 | 0.579 |
| Day 30 | 20 (25.6) | 3N, 1F | 0.125 | 21 (26.9) | 2N, 2F | 0.070 | 0.94 | 0.46–1.91 | 0.856 |
| Day 60 | 22 (28.2) | 2N | 0.375 | 25 (32.1) | 3N, 2NF,1F | 0.500 | 0.83 | 0.42–1.65 | 0.601 |
| Day 120 | 25 (32.1) | 3N | 1.000 | 25 (32.1) | NC | 0.500 | 1.00 | 0.51–1.96 | 1.000 |
| Day 180 | 25 (32.1) | 1N, 1F | ref | 27 (34.6) | 2N | ref | 0.89 | 0.46–1.73 | 0.734 |

[a] Within group comparison using day 180 as a reference using the McNemar test.

[b] Comparing the point abstinence rate between groups using the logistic regression.

N stands for new cases who could stop smoking at least 7 days before the visit.

F stands for participants who returned to smoking.

NF stands for participants who returned to smoking and could be abstinent later.

NC stands for no changes.

**Table 4. Comparisons of number of cigarettes smoked per day between the intervention and control groups on days 0, 7, 14, 30, 60, 120 and 180.**

| Visit | Intervention group (n = 78) | | p-value[a] | Control group (n = 78) | | p-value[a] | p-value[b] |
|---|---|---|---|---|---|---|---|
| | Number of cigarettes smoked per day Mean ±SD | 95%CI | | Number of cigarettes smoked per day Mean ±SD | 95% CI | | |
| Day 0 | 12.0 ± 7.0 | 10.40–13.57 | <0.001 | 12.3 ± 8.4 | 10.46–14.24 | <0.001 | 0.767 |
| Day 7 | 8.3 ± 7.6 | 6.55–9.96 | <0.001 | 6.9 ± 7.2 | 5.30–8.52 | <0.001 | 0.256 |
| Day 14 | 5.8 ± 5.9 | 4.44–7.10 | <0.001 | 5.7 ± 6.4 | 4.29–7.14 | <0.001 | 0.961 |
| Day 30 | 4.3 ± 4.2 | 3.32–5.21 | <0.001 | 3.0 ± 3.7 | 2.19–3.86 | 0.029 | 0.069 |
| Day 60 | 3.4 ± 5.7 | 2.12–4.70 | 0.012 | 2.7 ± 4.9 | 1.69–3.90 | 0.119 | 0.450 |
| Day 120 | 3.0 ± 4.1 | 2.07–3.93 | 0.003 | 2.6 ± 3.5 | 1.84–3.44 | 0.051 | 0.541 |
| Day 180 | 1.8 ± 2.9 | 1.13–2.46 | ref | 1.9 ± 3.2 | 1.14–2.60 | ref | 0.880 |

[a] Within group comparison using mixed models (linear: Heterogeneous First-Order Autoregressive). Day 180 was a reference.

[b] between groups comparison using linear regression.

25% in the intervention group (using the NRT2Quit app) and 8% in the control group (p = 0.19) [19]. A study by Nomura et al. in Japan showed no significant difference in continuous abstinence rates between telemedicine counseling with CureApp and face-to-face clinical visits with CureApp (81.0% vs. 78.9%) [28]. Another double-blind randomized controlled trial study by Bricker et al. compared two apps (SmartQuit and QuitGuide) over two months. This study showed quit rates of 13% in SmartQuit and 8% in QuitGuide (p > 0.05) [18]. However, a study on physicians and CureApp by Masaki et al. in Japan showed a significant difference between the intervention group using CureApp and a control group using a control-app (63.9% vs. 50.5%) [29].

The magnitude of the 7-day point abstinence rate in this study was similar to that of other pharmacist-led smoking cessation programs. A study in Qatar by Hajj et al. evaluated smoking cessation rates provided by pharmacists at 6 months and found a rate of 27.0% [30]. Gong et al. conducted an RCT study with participants who received pharmacist-provided telephone counseling and medications that revealed a 42.3% 1-week point abstinence rate at week 12, which was higher than the usual care rate of 38.2% (p > 0.05) [31]. A single-arm study by Iacoviello et al. in the US using the Clickotine app for 8 weeks, showed a self-reported

**Table 5. Comparisons of carbon monoxide (CO) levels between the intervention and control groups on days 0, 7, 14, 30, 60, 120, and 180.**

| Visit | Intervention group (n = 78) | p-value[a] | Control group (n = 78) | p-value[a] | OR | 95%CI | p-value[b] |
|---|---|---|---|---|---|---|---|
| | CO <7 ppm No (%) | | CO <7 ppm No (%) | | | | |
| Day 0 | 24 (30.8) | 0.017 | 24 (30.8) | 0.001 | 1.00 | 0.51–1.97 | 1.000 |
| Day 7 | 30 (38.5) | 0.210 | 32 (41.0) | 0.057 | 0.90 | 0.47–1.71 | 0.744 |
| Day 14 | 29 (37.2) | 0.039 | 36 (46.2) | 0.289 | 0.69 | 0.36–1.31 | 0.256 |
| Day 30 | 33 (42.3) | 0.375 | 38 (48.7) | 0.625 | 0.77 | 0.41–1.45 | 0.422 |
| Day 60 | 36 (46.2) | 1.000 | 38 (48.7) | 0.500 | 0.90 | 0.48–1.69 | 0.748 |
| Day 120 | 32 (41.0) | 0.125 | 38 (48.7) | 0.500 | 0.73 | 0.39–1.38 | 0.335 |
| Day 180 | 36 (46.2) | ref | 40 (51.3) | ref | 0.81 | 0.43–1.53 | 0.522 |

CO stands for exhaled carbon monoxide, ppm stands for part per million.

[b] Comparing exhaled CO between groups using the logistic regression.

[a] within group comparison using day 0 as a reference using the McNemar test.

**Table 6. Adverse drug reactions (ADR) reported in the intervention and control groups.**

| Adverse events | Intervention group (n = 40) No (%) | Control group (n = 44) No (%) |
|---|---|---|
| Total patients who had adverse events | 11 (27.5) | 16 (36.4) |
| Total events | 19 (100) | 24 (100) |
| Dry mouth, dry throat | 9 (47.4) | 12 (50.0) |
| Drowsiness with/without dry mouth | 1 (5.3) | 5 (20.8) |
| Dizziness | 0 (0.0) | 1 (4.2) |
| Chest discomfort | 1 (5.3) | 0 (0.0) |
| Palpitation | 2 (10.5) | 0 (0.0) |
| Vomiting | 2 (10.5) | 0 (0.0) |
| Nausea with frequently urination | 1 (5.3) | 0 (0.0) |
| Flatulence | 1 (5.3) | 0 (0.0) |
| GERD | 0 (0.0) | 1 (4.2) |
| Burning throat with/without GI discomfort | 1 (5.3) | 2 (8.3) |
| Numbness at mouth | 1 (5.3) | 2 (8.3) |
| Unable to hear well | 0 (0.0) | 1 (4.2) |

abstinence rate of 26.2% [20]. However, another study by Bricker et al. showed a lower quit rate (13% in SmartQuit and 8% in QuitGuide) when compared to our study [18]. Overall, this study with PharmQuit demonstrated better or similar cessation rates to those of these two previous studies.

Although most participants in the intervention group liked PharmQuit, it was accessed less frequently over time. Some participants explained that this was due to limited internet access. One participant complained that there were too many messages sent (twice per day), which may be counterproductive. In a study by Do et al., it was revealed that daily texts are less effective than weekly texts [32]. One participant uninstalled PharmQuit because of the limited memory on his phone, while another forgot to update the daily record because there was no feedback from his pharmacist. Some participants who were able to quit smoking stopped using PharmQuit after quitting. At the start of recruitment, the login feature malfunctioned, delaying recruitment by a week, which may have left a poor impression on the treatment group. These factors may explain the low rate of active users of PharmQuit, even though all participants in the intervention group were trained to use the app by their pharmacists at the start of the program. However, when considering that the aim of PharmQuit was to help participants adhere to the program with an easy-to-use app, the aim was achieved.

Another explanation for the results was the high loss to follow-up rate in both groups. Although the pharmacists reminded the participants about the appointments and made calls, most of them could not be reached by phone. This may have reduced the power of the study to show significant differences in smoking cessation rates between the two groups. One participant in the intervention group was diagnosed with cancer and decided to stop using the app. Other studies have shown that special characteristics, such as swearing to complete the study [28], much better compensation (90 US dollars per visit) [29], and effective medications (such as varenicline, bupropion, nicotine patch) [28–31] helped to increase app engagement and quit rates. Our study was performed ethically, and the participants could choose to leave the study at any time. A small compensation was provided, and any needed medications were provided free of charge. The most effective medication for smoking cessation in our study was nicotine gum, which was shown to be less effective than other medications in previous studies

**Table 7. Adherence rate to the smoking cessation program in the intervention and control groups.**

| Day | Adherence rate | | p-value[a] |
|---|---|---|---|
| | Intervention group (n = 78) Mean ± SD | Control group (n = 78) Mean ± SD | |
| Number of visits | 3.1±1.7 | 3.0±1.8 | 0.721 |
| Number of days adhered to the program | 74.3±76.1 | 60.1±70.8 | 0.316 |

[a] Comparing between groups by Mann-Whitney U test.

[33]. As only 51–56% of participants received medications, participants who did not receive medications may have thought that follow-up was not necessary or worthwhile.

This study has some limitations. A substantial number of participants had missing data during the follow-up visits. The compensation may not have been a sufficient incentive to convince people to join and complete the study. Although participants who were engaged in other cessation programs during recruitment were excluded, no follow-up was conducted to determine if they had started another cessation program during the study. The interface of PharmQuit is in Thai, so the use of PharmQuit outside of Thailand would be limited to countries with Thai as a first or second language.

The strength of the study was that it was a multicenter study that used a randomized control design and had a long follow-up duration. Some of the community pharmacists in this study had experience serving at a university with a smoke-free campus policy [34].

## Conclusions

The results showed the benefit that community pharmacists provide in helping smokers quit smoking. PharmQuit showed better or similar quit rates to those of some other studies. Although PharmQuit was not more effective than pharmacist counseling, it may help pharmacists achieve better adherence to smoking cessation programs. Thus, the use of a mobile app is one option for smokers in larger clinical trials with an aim of smoking cessation.

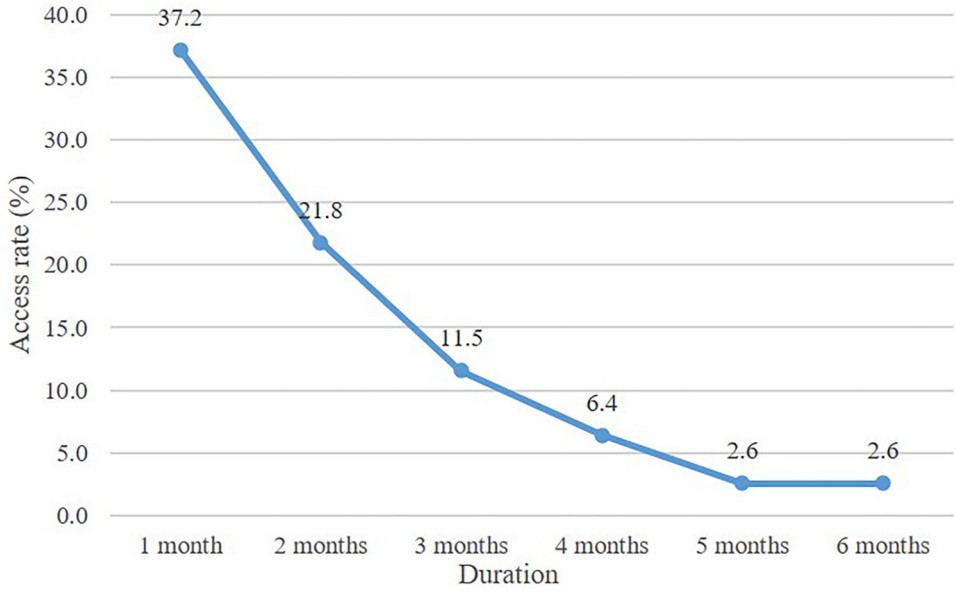

**Fig 2. Access rate to PharmQuit in the intervention group within the 6-month follow-up (n = 78).**

**Table 8. Satisfaction score (on a scale of 1–5) to PharmQuit in the intervention group at 6 months.**

| PharmQuit | Median (range) (n = 14) |
|---|---|
| Dimension 1: Objective to quit smoking | |
| 1. You are satisfied with PharmQuit in helping you to keep service schedules. | 4.0 (2) |
| 2. You are satisfied with the progress feature. | 4.5 (2) |
| 3. You are satisfied with the encouragement received | 4.0 (2) |
| 4. You are satisfied with question and answer section. | 4.0 (2) |
| 5. You are satisfied that PharmQuit has helped you quit or reduce the number of cigarettes smoked. | 4.0 (2) |
| Dimension 2: Scope of application | |
| 6. You are satisfied with the number of functions. | 4.0 (2) |
| 7. You are satisfied with interactive functions between a pharmacist and other smokers. | 4.0 (3) |
| 8. You are satisfied with the ease of inputting your personal information. | 4.0 (2) |
| 9. You are satisfied with the privacy of your information. | 4.0 (3) |
| Dimension 3: Format and interactive between PharmQuit and the user | |
| 10. You are satisfied with the daily encouraging messages and reminders. | 4.0 (3) |
| 11. You are satisfied with the response speed of the application. | 4.0 (3) |
| 12. You are satisfied with humorous and interesting features. | 4.0 (3) |
| 13. You are satisfied with the challenging and attractive interactive features. | 4.0 (3) |
| Dimension 4: Design | |
| 14. You are satisfied with characteristics of the app. | 4.0 (3) |
| 15. You are satisfied with the amount of information on each screen. | 4.0 (4) |
| 16. You are satisfied with the sequence of each group of functions. | 4.0 (3) |
| 17. You are satisfied with the convenience and ease of use of PharmQuit. | 4.0 (1) |
| Dimension 5: Appearance | |
| 18. You are satisfied with attractiveness and usability of the app. | 4.5(3) |
| 19. You are satisfied with the font and background color. | 4.0 (3) |
| 20. You are satisfied with the font and font size. | 5.0 (4) |
| 21. You are satisfied with beautiful and attractive pictures used. | 4.0 (2) |

# Supporting information

**S1 File. CONSORT 2010 checklist of information to include when reporting a randomized trial.**
(PDF)

**S2 File. Research protocol as approved by the ethics committee.**
(PDF)

**S3 File. English and Thai questionnaire of satisfaction with PharmQuit.**
(PDF)

# Acknowledgments

We are grateful for the contributions of Asst. Prof. Krongjit Vathesatogkit, RN, Asst. Prof. Theerapong Seesin, Pharm.D, BCPs, Asst. Prof. Dr. Saithip Suttiruksa, Charnkiat Phianchana, Pharm.D., high school students in Kaengkhro Industrial and Community Education College, and the team of programmers at the Faculty of Informatics for PharmQuit development. We wish to thank Prof. P.T. Thomas for his kind comments on the methodology and proofreading. We also wish to thank the native English speaker, Mr. Douglas Olson, for proofreading.

We are thankful to the seven community pharmacies, batch 14 Pharm.D. Students from Maha-sarakham University, health care providers in Takhonyang Sub-district, and Khamrieng Sub-district for helping to recruit participants and being extremely supportive in this study.

## Author Contributions

**Conceptualization:** Narong Asayut, Phayom Sookaneknun Olson, Juntip Kanjanasilp, Preut Thanarat, Bhattaraporn Senkraigul, Chuthathip Sittisarn, Suratsawatee Suksawat.

**Data curation:** Narong Asayut, Phayom Sookaneknun Olson, Chuthathip Sittisarn, Suratsa-watee Suksawat.

**Formal analysis:** Narong Asayut, Chuthathip Sittisarn, Suratsawatee Suksawat.

**Funding acquisition:** Phayom Sookaneknun Olson.

**Investigation:** Narong Asayut, Phayom Sookaneknun Olson, Juntip Kanjanasilp, Chuthathip Sittisarn, Suratsawatee Suksawat.

**Methodology:** Narong Asayut, Phayom Sookaneknun Olson, Juntip Kanjanasilp, Preut Tha-narat, Chuthathip Sittisarn, Suratsawatee Suksawat.

**Project administration:** Narong Asayut.

**Resources:** Phayom Sookaneknun Olson.

**Software:** Narong Asayut, Phayom Sookaneknun Olson, Juntip Kanjanasilp, Preut Thanarat, Bhattaraporn Senkraigul.

**Supervision:** Narong Asayut, Phayom Sookaneknun Olson, Juntip Kanjanasilp.

**Validation:** Narong Asayut, Phayom Sookaneknun Olson, Juntip Kanjanasilp, Preut Thanarat, Bhattaraporn Senkraigul.

**Visualization:** Phayom Sookaneknun Olson, Juntip Kanjanasilp.

**Writing – original draft:** Narong Asayut.

**Writing – review & editing:** Phayom Sookaneknun Olson, Juntip Kanjanasilp, Preut Tha-narat, Bhattaraporn Senkraigul, Chuthathip Sittisarn, Suratsawatee Suksawat.

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
