## [Decision Letter · Decision Letter 0]

26 Jul 2021

PONE-D-21-17146

A community pharmacist-led smoking cessation intervention using a smartphone app (PharmQuit): A randomized controlled trial

PLOS ONE

Dear Dr. Olson,

Thank you for submitting your manuscript to PLOS ONE. After careful consideration, we feel that it has merit but does not fully meet PLOS ONE’s publication criteria as it currently stands. Therefore, we invite you to submit a revised version of the manuscript that addresses the points raised during the review process.

We look forward to receiving your revised manuscript.

Kind regards,

Shahrad Taheri

Academic Editor

PLOS ONE

Furthermore, please provide additional information regarding how participants were recruited for the study.

6. Thank you for stating the following in the Funding Section of your manuscript:

“This study was supported by the Health Security Office region 7 Khon Kaen [PO, grant number 60/B/00293, 2017], Health Promotion Smoke Free Pharmacy Network, Community Pharmacy Foundation, and Thai Health Promotion Foundation [PO, grant number 63-05-009, 2020]. The funders had no role in study design, data collection and analysis, decision to publish, or preparation of the manuscript”

Funding information should not appear in the Funding section or other areas of your manuscript. We will only publish funding information present in the Funding Statement section of the online submission form.

 “This study was supported by the Health Security Office region 7 Khon Kaen [PO, grant number 60/B/00293, 2017], Health Promotion Smoke Free Pharmacy Network, Community Pharmacy Foundation, and Thai Health Promotion Foundation [PO, grant number 63-05-009, 2020]. The funders had no role in study design, data collection and analysis, decision to publish, or preparation of the manuscript.”

Reviewers' comments:

Reviewer's Responses to Questions

**Comments to the Author**

1. Is the manuscript technically sound, and do the data support the conclusions?

Reviewer #1: Partly

Reviewer #2: Partly

2. Has the statistical analysis been performed appropriately and rigorously? 

Reviewer #1: No

Reviewer #2: I Don't Know

3. Have the authors made all data underlying the findings in their manuscript fully available?

Reviewer #1: No

Reviewer #2: Yes

4. Is the manuscript presented in an intelligible fashion and written in standard English?

Reviewer #1: No

Reviewer #2: Yes

5. Review Comments to the Author

Reviewer #1: This is a very interesting randomised controlled studying evaluating the efficacy of an app called PharmQuit to help with smoking cessation in Thailand. Using Pharmacists to carry out the interventions.

Well done to team for carrying out such important research.

They are some comments worth mentioning for the authors attention.

1) Line 103, can the aims be clear in terms of what the primary objectives, secondary objectives are.

2) Under design, include that it was multi-centre/site since recruitment occurred in 7 community pharmacies in 3 provinces

3) The sample size calculation is incomplete, what was the quit rate (i.e the assumed quit rate together with what the ‘improvement’ was planned to be?

4) In addition if the quit rate was to be estimated at 6 months follow-up, this implies that 6 month is the primary timepoint. This needs to be clear especially when mentioning the aims.

5) Randomization allocation ratio, I assume it was 1:1, this needs to be explicitly stated.

6) Although this an open-label study, allocation concealment needs to be mentioned and explicitly stated who was not blinded.

7) For the randomization procedure of the pharmacist checking the randomisation list, what was the assurance there was no selection bias. Was an external check/validation carried out. For an example how do you know say the next group allocation was control group but then the Pharmacist then allocated the person to intervention?

8) It would benefit clarity of the manuscript if there was a separate heading “Outcomes”. That way to make it explicit that outcome is binary, misleading to use rate as this implies a count when infact it’s a logistic model that has been fitted. Its also confusing, as it looks like there were two primary outcomes i.e. quit rate and number of cigarettes smoked perday. However the sample size only shows calculations for quit rate. What about the effect size for number of cigarettes smoked per day. And whether this is a co-primary end point, this needs to elaborated in more detail.

9) Also how was the satisfaction scored? I.e. total score for each dimension? And with such small numbers of 14 people completing the questionnaire, is it best to give a median instead with range?

10) It would help if the Title for Table 6 also includes that this was restricted to intervention group only. You probably also need a ‘missing’ summary as well.

11) “We randomized a total of 156 participants to the intervention group or the control group.” Best if this also includes numbers randomised to each arm.

12) In terms of the statistical analysis, the data structure was repeated measurements of some outcomes, should this have incorporated a mixed effects model approach.

13) how was missing data handled?

14) Any statistical analysis plan written? and presumably signed off prior to any data analysis.

Items with CONSORT checklist

1) Outcomes (Item 6a)

Completely defined pre-specified primary outcome measure including how and when it was assessed

Is it clear (1) what the primary outcome is (usually the one used in the sample size calculation), (2) how it was measured (if relevant; e.g. which score used), (3) at what time point, and (4) what the analysis metric was (e.g. change from baseline, final value)?

Not adequately defined. Primary timepoint not specified as data is being collected at different timepoints.

2) Sample size (Item 7a)

How sample size was determined

Is there a clear description of how the sample size was determined, including (1) the estimated outcomes in each group; (2) the α (type I) error level; (3) the statistical power (or the β (type II) error level); and (4) for continuous outcomes, the standard deviation of the measurements?

Sample size not adequately stated as well as what the effect size is. Cannot be replicated.

3) Sequence generation (Item 8a)

Method used to generate random allocation sequence

Does the description make it clear if the "assigned intervention is determined by a chance process and cannot be predicted"?

4) Allocation concealment (Item 9)

Mechanism used to implement random allocation sequence (such as sequentially numbered containers), describing any steps taken to conceal the sequence until interventions were assigned

Is it clear how the care provider enrolling participants was made ignorant of the next assignment in the sequence (different from blinding)? Possible methods can rely on centralised or "third-party" assignment (i.e., use of a central telephone randomisation system, automated assignment system, sealed containers).

Not clearly stated in the manuscript. The method of allocation is unblinded.

5) Blinding (Item 11a)

If done, who was blinded after assignment to interventions (for example, participants, care providers, those assessing outcomes)

Is it clear if (1) healthcare providers, (2) patients, and (3) outcome assessors are blinded to the intervention? General terms such as "double-blind" without further specifications should be avoided.

Open-label, but it would be good to state if person doing the analysis was blinded to the data.

6) Outcomes and estimation (Item 17a/b)

For the primary outcome, results for each group, and the estimated effect size and its precision (such as 95% confidence intervals)

Is the estimated effect size and its precision (such as standard deviation or 95% confidence intervals) for each treatment arm reported? When the primary outcome is binary, both the relative effect (risk ratio, relative risk) or odds ratio) and the absolute effect (risk difference) should be reported with confidence intervals.

Just the odds ratio (95% CI) has been reported.

7) Harms (Items 19)

All important harms or unintended effects in each group

Is the number of affected persons in each group, the severity grade (if relevant) and the absolute risk (e.g. frequency of incidence) reported? Are the number of serious, life threatening events and deaths reported? If no adverse event occurred this should be clearly stated.

Safety outcomes not mentioned.

8) Registration (Item 23)

Registration number and name of trial registry

Is the registry and the registration number reported? If the trial was not registered, it should be explained why.

Yes,

9) Protocol (Item 24)

Where trial protocol can be accessed

Is it stated where the trial protocol can be assessed (e.g. published, supplementary file, repository, directly from author, confidential and therefore not available)?

Yes, in supplementary

10) Funding (Item 25)

Sources of funding and other support (such as supply of drugs) and role of funders

Are (1) the funding sources, and (2) the role of the funder(s) described?

Yes.

Reviewer #2: The authors should be commended for running an overall well designed study, and writing a generally high quality paper.

Unfortunately I do have one major concern about how the ITT protocol was followed - the authors state an ITT protocol was developed, but the results are a bit unclear and seem to indicate that this was perhaps not the case? Clarification is urgently needed.

Apart from this one issue I am more than happy to recommend publication, but this issue must be adequately addressed as a matter of urgency, as it brings the major part of the results/conclusion of the paper into question. More details provided below.

Major comments:

208: "There were a total of 7 participants who relapsed after quitting". This is a completely incorrect statement. You said you used an intention to treat methodology. Therefore every participant who dropped out of the study (the vast majority!) is assumed to have resumed smoking! This must be qualified very clearly! This mistake makes me wonder if your other results are genuinely calculated on an intention to treat basis? i.e. in Table 3, you say n=78 in each group but then your quit rate goes up over time - I suspect you have not included the drop-outs (as relapsers) in this data. This same comment can potentially be made for others of the reported results.

Minor comments:

Design: please include the $ symbol and report to 2 decimal places the currency information at lines 111 and 112

Ethical approval: failure to register the clinical trial prior to enrolment is a fairly big mistake under the current clinical trials rules, however, it is only very recently that a trial like this would have been considered a clinical trial at all. The excuse about the requirement not being mentioned "in the graduate program" is a bit confusing though (the reader doesn't know anything about your "graduate program" or how it relates to this mistake). I suggest replacing this statement with something like "The researchers were unaware of the requirement to register the clinical trial prior to the enrolment of participants, but amendments have been made to our institutional training materials to reduce the likelihood of this error occurring in the future." If possible, it may also be valuable to state that the HREC was made aware of this mistake and judged it to be low risk, or something to that effect.

Lines 130-131: Is there a reference to support the estimated drop-out rate of 15%?

Line 134: Fagerstorm => Fagerström

Line 148: The whole URL should be hyperlinked, not just the inner part.

Line 166: "usual pharmacy services on smoking cessation and medications" => It is important to state explicitly what these usual services are, since practice in various regions differs significantly. Most important is the conditions under which participants would gain access to NRT, and the frequency of counselling/assessments - I assume that this is the same as in the intervention group, but this must be made explicitly clear. It may be beneficial to instead report the common characteristics of treatments in both groups, then state how the intervention and control groups differed (presumably the only difference was access to PharmQuit). As it is currently written, it seems like the intervention group gets WAY more care than the control group but I do not think this is correct (and it would be a major design flaw if true).

Line 172-173: "Quit rate was determined by counting the number of visits with the pharmacist since quitting" this doesn't make sense.

Line 177 "User’s experience theory ": a reference should be included.

Line 186: " using the logistic regression" => "using logistic regression"

Line 186-187: "Data for participants with missing data were assumed to be the same as their last visit information." -> this must be justified. Was it the policy of the recording pharmacists to leave fields blank if they were unchanged? Missing data is not normally treated this way.

Line 187: " SPSS version 19" It's a little unusual to use such an ancient version of SPSS, but probably not of an importance given the statistical analyses are quite straightforward.

Results

Line 190-191: your completion rate makes me think your estimated drop-out rate of 15% was extremely naive? Unless you meant 15% drop-out per visit? Some follow-up with participants who were "unable to be contacted" might be useful to figure out what their reasons for dropping out were (presumably they relapsed and were no longer interested in quitting, but it would be nice to get some data).

196: "More medications were used by participants in the control group than in the intervention group" - this is clearly false when looking at the table. There is no significant difference, and there doesn't seem to be a huge clinical difference between 51% and 56%.

196-197: "FTND in the control group was lower in the high nicotine addiction group than in the intervention group" - This sentence is difficult to understand, as we have not been introduced to the concept of "high nicotine dependence" groups before now. Upon examination of the table it makes sense, but it is a clumsy sentence.

Discussion:

287: "this good" => "this could"

Discussion re: other quit smoking apps fails to highlight that each study is using a very different protocol and this is likely the largest reason for the large discrepancies in outcomes. i.e. if you don't have face to face interviews drop out rates are higher etc.

6. PLOS authors have the option to publish the peer review history of their article (what does this mean?). If published, this will include your full peer review and any attached files.

Reviewer #1: No

Reviewer #2: No

---

## [Author Response · Author response to Decision Letter 0]

20 Oct 2021

Thank you very much for all comments to help us improve our manuscript. We clarified and changed the outcome measures and analysis as recommended. We made all other suggested changes as well.

---

## [Decision Letter · Decision Letter 1]

3 Mar 2022

A community pharmacist-led smoking cessation intervention using a smartphone app (PharmQuit): A randomized controlled trial

PONE-D-21-17146R1

Dear Dr. Olson,

We’re pleased to inform you that your manuscript has been judged scientifically suitable for publication and will be formally accepted for publication once it meets all outstanding technical requirements.

Kind regards,

Shahrad Taheri

Academic Editor

PLOS ONE

Additional Editor Comments (optional):

Reviewers' comments:

Reviewer's Responses to Questions

**Comments to the Author**

1. If the authors have adequately addressed your comments raised in a previous round of review and you feel that this manuscript is now acceptable for publication, you may indicate that here to bypass the “Comments to the Author” section, enter your conflict of interest statement in the “Confidential to Editor” section, and submit your "Accept" recommendation.

Reviewer #1: All comments have been addressed

Reviewer #2: (No Response)

2. Is the manuscript technically sound, and do the data support the conclusions?

Reviewer #1: Yes

Reviewer #2: Yes

3. Has the statistical analysis been performed appropriately and rigorously? 

Reviewer #1: Yes

Reviewer #2: Yes

4. Have the authors made all data underlying the findings in their manuscript fully available?

Reviewer #1: No

Reviewer #2: Yes

5. Is the manuscript presented in an intelligible fashion and written in standard English?

Reviewer #1: Yes

Reviewer #2: Yes

6. Review Comments to the Author

Reviewer #1: (No Response)

Reviewer #2: Thanks for submitting your revision. The paper is significantly improved.

I would note that a limitation of the design is that your satisfaction survey was only given to participants who completed the whole program - this may introduce a selection bias (presumably participants who were unsatisfied would drop out, and participants who were satisfied would remain).

7. PLOS authors have the option to publish the peer review history of their article (what does this mean?). If published, this will include your full peer review and any attached files.

Reviewer #1: No

Reviewer #2: **Yes: **Dr Ivan Karl Bindoff

---

## [Editor Report · Acceptance letter]

8 Mar 2022

PONE-D-21-17146R1 

A community pharmacist-led smoking cessation intervention using a smartphone app (PharmQuit): A randomized controlled trial 

Dear Dr. Olson:

I'm pleased to inform you that your manuscript has been deemed suitable for publication in PLOS ONE. Congratulations! Your manuscript is now with our production department. 

Kind regards, 

on behalf of

Dr. Shahrad Taheri 

Academic Editor

PLOS ONE